# LatentCBF: A Control Barrier Function in Latent Space for Safe Control

## Abstract

Safe control is crucial for safety-critical autonomous systems that are deployed in dynamic and uncertain environments. Quadratic-programming-control-barrier-function (QP-CBF) is becoming a popular tool for safe controller synthesis. Traditional QP-CBF relies on explicit knowledge of the system dynamics and access to all states, which are not always available in practice. We propose LatentCBF (LCBF), a control barrier function defined in the latent space, which only needs an agent's observations, not full states. The transformation from observations to latent space is established by a Lipschitz network-based AutoEncoder. In addition, the system dynamics and control barrier functions are all learned in the latent space. We demonstrate the efficiency, safety, and robustness of LCBFs in simulation for quadrotors and cars.

## 1 INTRODUCTION

Many safety-critical robotic systems are deployed in dynamic and uncertain environments, such as autonomous cars, delivery drones, etc. To provably guarantee the safe control of these robots is challenging. Quadratic programming-control-barrier-function (QP-CBF), which solves an optimization problem with safety as a hard constraint online, is becoming a popular tool for safe controller synthesis. Thanks to the forward invariant set property, the trajectories of a dynamic system can be assured to stay in a safe set over an infinite time horizon Ames et al. (2014; 2016). The primary challenge in existing research lies in the construction of a CBF. Presently, numerous works depend on manually crafting CBFs, which proves effective for simple systems. However, when additional factors such as the relative degree of nonlinear dynamics Xiao & Belta (2021) and constraints arising from control limits come into play Liu et al. (2023), the task of manual design becomes increasingly complex. In light of these complexities, a data-driven approach for automating CBF synthesis emerges as a promising alternative.

Another significant limitation of existing works is that the theoretical safety assurances provided by CBFs are based on the presumption that all states are accessible within the dynamic model and the environment Chen et al. (2023); Dean et al. (2021). This creates a significant gap when applying CBF theory in practical scenarios, especially in the context of many robotic systems that rely on sensors or images as inputs, where obtaining ground truth states is often impossible.

This work presents a unified control schema that employs a data-driven CBF, neural dynamics, and a reinforcement learning policy all in a unified latent space. The data-driven CBF utilizes a Lipschitz neural network (NN) Anil et al. (2019), effectively eliminating the need for manual design. A reinforcement learning schema is adopted to adaptively compensate uncertainty of a dynamic model.

The contributions of our paper can be summarized as follows:

- Utilize Lipschitz networks for an end-to-end control problem and demonstrate its robust and utility in our novel framework.
- To the best of our knowledge, we are the first to propose a unified control architecture defined in latent space, which employs a neural CBF, a control affine dynamics model and a neural policy.

The rest of this paper is organized as follows. Related work is discussed in Sec. 2. We elaborate on the LatentCBF in Sec. 3. The results are in Sec. 4. Conclusions and future work are in Sec. 5.

## 2 RELATED WORKS

We will briefly review the most important related works in two main categories:

1) *Barrier functions for safety certificates*: Dawson et al. (2022a),Zhang et al. (2023) Dawson et al. (2023b) Tan et al. (2023) and Lindemann et al. (2021) proposes a neural barrier function to represent the safe set. It is only used for verifying the state, unlike CBF; safety certificates are not used for controller synthesis.

2) *Barrier functions for controller synthesis*: Dawson et al. (2022b); Xiao et al. (2022; 2021) uses NN-based barrier functions. However, the dynamics models are known. Xiao et al. (2021) proposes *BarrierNet*, which requires an available dynamics model and pre-defined Hessian and linear cost matrices for the QP. Our approach can learn system dynamics and optimal Hessian and linear cost matrices for the QP. Dawson et al. (2022b) proposes *rCBLF-QP*, which learns a Lyapunov function for the QP via exploration. The major limitation is that the typical definition of a barrier function requires state information that is not generally available in real-world scenarios. Our approach only requires the observation space. **We compare our approach with *rCBLF-QP* and *BarrierNet* in our experiments**.

3) *Differentiable QP solver*: Amos & Kolter (2017) formulates how a QP-based optimization can be added as a differentiable layer for the last layer of a NN. It proposes to learn the Hessian and linear cost matrices with a cost function by backpropagating through the network. Xiao et al. (2021) is a similar work but uses a high-order CBF (HOCBF). Its limitation, again, is the requirement of fully available state information.

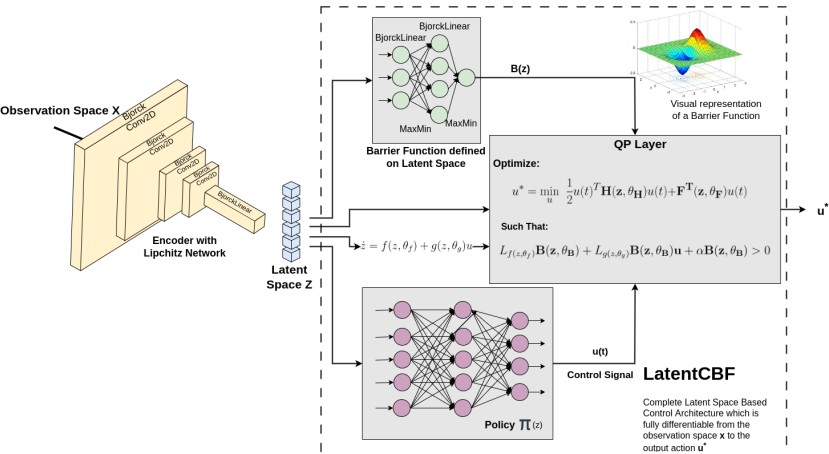

Figure 1: The illustration of the whole proposed framework. A simple barrier function is visualized with two features i.e., the coordinates as an example.

## 3 LATENT CONTROL BARRIER FUNCTION

### 3.1 PRELIMINARY

The basis of our formulation is the latent space learned using a *Lipschitz AutoEncoder*. We define the following *encoder* $E(.)$ and *decoder* $D(.)$:

$$
\begin{aligned}
z &= E(x) \\
\hat{x} &= D(z)
\end{aligned}
\tag{1}
$$

where $E : X \to Z$, $D : Z \to \hat{X}$. The original observation space $X$ and $\hat{X}$ are both in the Euclidean space $\in \mathbb{R}^D$. The latent space $Z \in \mathbb{R}^d$. The latent space is used by individual modules of our framework, which will be explained in the upcoming sections. Our encoder is based on a Lipschitz NN:

$$|E(x_1) - E(x_2)| \leq C|x_1 - x_2|$$
$$|z_1 - z_2| \leq C|x_1 - x_2| \tag{2}$$

where $z_1, z_2 \in Z$, $x_1, x_2 \in X$, and $C$ is a Lipschitz constant. More details on the architecture, the construction and the advantages of our autoencoder are given in Sec. 3.3.

We assume a general nonlinear control-affine system in the learned latent space as follows:

$$\dot{z} = f(z, \theta_f) + g(z, \theta_g)u. \tag{3}$$

where $f : \mathbb{R}^d \to \mathbb{R}^d$, $g : \mathbb{R}^d \to \mathbb{R}^{d \times q}$, and $u \in \mathbb{R}^q$. $f(.)$ and $g(.)$ are respectively parameterized by $\theta_f$ and $\theta_g$ using NNs and learned in a pipeline explained in Sec. 3.4.

A control barrier function in the latent space is defined using a Lipschitz NN $B(z, \theta_B) : \mathbb{R}^d \to \mathbb{R}$. The safety is guaranteed by establishing the following constraint Ames et al. (2016):

$$\dot{B}(z) > -\alpha B(z) \tag{4}$$

where $\alpha$ is a Lipschitz class $\mathcal{K}$ function, which can be chosen as a user-defined constant as its simplest form Ames et al. (2014). Eq. 4 is expanded as follows:

$$L_{f(z,\theta_f)}B(z) + L_{g(z,\theta_g)}B(z)u(t) > -\alpha B(z) \tag{5}$$

where $L_{f(z,\theta_f)}B(z) = \frac{dB(z,\theta_B)}{dz}f(z, \theta_f)$ and $L_{g(z,\theta_g)}B(z) = \frac{dB(z,\theta_B)}{dz}g(z, \theta_g)$ are Lie derivatives.

Integrating the constraint Eq. 5 into a quadratic program, a QP-CBF controller synthesis can be obtained by solving the following QP problem online:

$$\min_{u} \quad u^T H(z, \theta_H)u + u^T F(z, \theta_F)$$
$$\text{s.t. } L_{f(z,\theta_f)}B(z) + L_{g(z,\theta_g)}B(z)u + \alpha B(z) > 0 \tag{6}$$
$$u_{min} \leq u \leq u_{max}$$

where $u_{min}$ and $u_{max}$ are control limits. $H(.)$ and $F(.)$ are Hessian and linear cost matrices. $H(z, \theta_H) : \mathbb{R}^d \to \mathbb{R}^{q \times q}$ and $F(z, \theta_F) : \mathbb{R}^d \to \mathbb{R}^{q \times 1}$ are matrices and parameterized by $\theta_H$, and $\theta_F$ respectively. The architecture and training of $B(z, \theta_B)$, $H(z, \theta_H)$, and $F(z, \theta_F)$ will be explained in Sec. 3.5. The nominal controller in QP is a policy based on a shallow NN in the latent space, i.e., $\pi(z)$. There are two policies, each designated for an individual phase. 1) $\pi_{adapt}$: This is used to span over an ample amount of observation space to collect enough data samples for learning the latent space and the barrier function; 2) $\pi_{optimal}$: This is the optimal policy trained for the reward of the task in the environment.

$$\pi : \mathbb{R}^d \to \mathbb{R}^c; u = \pi(z, \theta_\pi) \tag{7}$$

More architectural details about the reward and training pipeline are given in Sec. 3.6.

## 3.2 OVERVIEW

Our proposed framework is illustrated in Fig. 1. The basic flow can be explained by the following steps: 1) The observation is embedded into the latent space using a Lipschitz encoder (see Eq. 1 and the blocks in yellow and blue in Fig. 1). 2) The latent space then is fed into the control pipeline consisting of a policy in Eq. 7, which outputs nominal control signal $u$ (see the purple NN). 3) The Lipschitz continuous CBF (see Eq. 4 and the green NN) and the dynamics in Eq. 3 are also defined in the latent space. 4) All of the aforementioned components are sent to the overall optimization in the QP layer (c.f., Eq. 6) that optimizes the control signal using the Hessian and linear cost matrices described in 3.1. The final optimized control action is $u^*$. Note that the pipeline only represents the forward pass; the following sections on individual modules explain their backpropagation. The gradients from all the modules are accumulated in the encoder for learning an optimal representation.

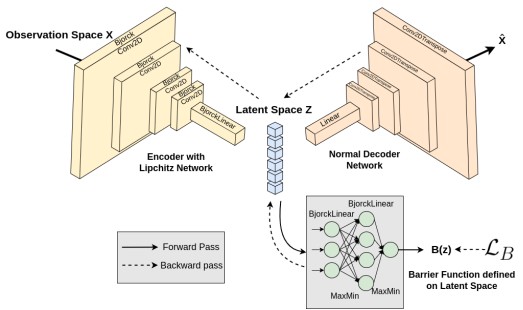

Figure 2: The illustration of our AutoEncoder design.

### 3.3 LEARNING LIPSCHITZ AUTOENCODER

The AutoEncoder is an essential component in our framework, since all other components are established in a latent space. The necessity of Lipschitz continuous autoencoder is to have barrier function that adheres to its original formulation given in Ames et al. (2016). It requires the barrier function to be a Lipschitz continuous function, but all the works on neural barrier function do not emphasize this constraint for the function. We conduct experiments in 4.2 and find out that our architecture that approximates the Lipschitz criterion is much more suitable for the task of defining a neural barrier function over a traditional network of similar architecture.

To obtain a Lipschitz network-based AutoEncoder Anil et al. (2019), we use an architecture similar to Ghifary et al. (2016), which has a classification head in the latent space to backpropagate on the encoder network. Such a design helps the autoencoder in domain adaptation in events of abrupt changes occurring due to training of the control pipeline. The proposed architecture is illustrated in Fig. 2. Following the dashed lines, the loss is backpropagated from all the networks using latent space and accumulates into the AutoEncoder.

The AutoEncoder network is trained by optimizing the following reconstruction loss given by $L_{recon} = \sum_{i=1}^{n_{batch}} |\hat{x}_i - x_i|$. The Autoencoder can be flexibly designed for any kind of input, e.g., simple measurement features or even images, by replacing a typical encoder for that modality with specific layers for the Lipschitz network. For example, to get a *Convolutional 2D* layer, we can replace it with the BjorckConv2D Anil et al. (2019), and for a *Linear Dense layer*, we can choose BjorckLinear Anil et al. (2019). We have limited options for activations, the most popular choice being *MaxMin* layer Anil et al. (2019).

### 3.4 LEARNING SYSTEM DYNAMICS

To learn $f(.)$ and $g(.)$ parameterized by NNs, we formulate a regression model. First, we expand Eq. 3 to get the following form:

$$\frac{d(z)}{dx}\frac{dx}{dt} = f(z, \theta_f) + g(z, \theta_g)u \tag{8}$$

where $\frac{d(z)}{dx}$ is the gradient computed in the backward pass of the encoder $E$ (see Eq. 1). For a given observation $x_0$, $\frac{dz}{dx}|_{x_0} = \frac{d(E(x))}{dx}|_{x_0}$ is easily calculated by backpropagation. $\frac{dx}{dt}|_{x_0}$ can be computed from the numeric difference between two consecutive observations. Hence the estimated values $\frac{dz}{dt}_{estim}$ will serve as labels for optimizing dynamics functions $f(z, \theta_f)$ and $g(z, \theta_g)$ in supervised learning approach using 8 with the following loss:

$$L_{dyn} = \sum_{i=1}^{n_{batch}} \left| \frac{dz}{dt}_{estim,i} - f(z, \theta_f)_i - g(z, \theta_g)_i u_i \right| \tag{9}$$

## 3.5 LEARNING BARRIER NETWORK

We construct a CBF parameterized by a Lipschitz network called *Barrier Network*. In our experiments, we use four Bjorck Linear Layers and the MaxMin activation function.

To learn the cost matrices $H(z, \theta_H)$ and $F(z, \theta_F)$ in Eq. 6, we use a policy distillation technique Robey et al. (2020), where the parameters are updated based on a loss function $l(.)$ measuring the similarity between the output of the latentCBF and expert trajectories from $\pi_{optimal}$ (see Eq. 7), which satisfies $Z_{traj} \in Z_{safe}$.

$$\theta = \arg\max_{\theta} \mathbb{E}[l(\pi_{optimal}(z), u^*)] \tag{10}$$

The gradient for this loss can be computed using the technique in Amos & Kolter (2017). It computes the Lagrangian of the qp formulation(we use our CBF QP formulation 6) and refactors it to a differentiable matrix. In our case, using the same technique, we get 11.

$$\begin{bmatrix} d_u \\ d_\lambda \end{bmatrix} = \begin{bmatrix} H & G^T D(\lambda^*) \\ G & D(Gu^* - h) \end{bmatrix}^{-1} \begin{bmatrix} (\frac{\partial l}{\partial u^*})^T \\ 0 \end{bmatrix} \tag{11}$$

where $G = -L_{g(z,\theta_g)}B(z)$, $h = L_{f(z,\theta_f)}B(z) + \alpha B(z)$, $D(\cdot)$ creates a diagonal matrix, and $\lambda$ is the dual variable for the QP formulation. Eq. 6 can be rewritten as follows:

$$\min_{u} \quad u^T H u + F^T u$$
$$\text{s.t.} \quad Gu < h \tag{12}$$

Once we obtain $d_u$ and $d_\lambda$ from Eq. 11, we can optimize parameterized $H(z, \theta_H)$ and $F(z, \theta_F)$ by

$$\nabla_H l = \frac{1}{2}(d_u u^{*T} + u^* d_u^T) \tag{13}$$
$$\nabla_F l = d_u \tag{14}$$

Now we discuss how to obtain an optimal barrier function. A straightforward mechanism to design the reward $r(x, u)$ is having a positive reward for reaching the goal, a negative reward for colliding with obstacles or completely deviating from the objective, and otherwise zero. This is generally the reward structure in the control problem, which is also consistent with our environments.

With such a reward design, we can have an annotating algorithm that identifies whether a particular state is safe or unsafe. For any $x \in X$, after the transformation to latent space under $E : X \to Z$, the agent's trajectory $X_{traj}$ becomes $Z_{traj}$ in the latent space. Hence for $z \sim Z_{traj}$

$$z \in \begin{cases} Z_{safe}, & \text{if } \mathbb{E}_{x(t) \sim X_{traj}}^{x_{term}} \left[\sum r(x(t), u(t))\right] - p_+ \geq 0 \\ Z_{unsafe}, & \text{if } \mathbb{E}_{x(t) \sim X_{traj}}^{x_{term}} \left[\sum r(x(t), u(t))\right] - p_- < 0 \end{cases} \tag{15}$$

where $\mathbb{E}_{x(t) \sim X_{traj}}^{x_{term}} [\sum r(x(t), u(t))]$ is the expected return or cumulative reward signal along the trajectory $X_{traj}$, $x_{term}$ is the terminal state of an episode or trajectory, $p_+$ is the minimum threshold to classify a trajectory safe, and $p_-$ is the maximum threshold to classify a trajectory to be unsafe. The values of $p_+$ and $p_-$ are hyperparameters. In our experiments we set $p_+ = 0.6 * r_{max}$ and $p_- = 0.2 * r_{min}$, where $r_{max} = \max_X r(x(t), u(t))$ and $r_{min} = \min_X r(x(t), u(t))$ can be obtained from the environment. With this information and data collected from the sampling phase using $\pi_{adapt}$, we have $x \in X_{traj}$. We have the following loss function for training $B(z, \theta_B)$:

$$L_B = \sum_{z \in Z_{safe}} |1 - B(z, \theta_B)| + \sum_{z \in Z_{unsafe}} |B(z, \theta_B) + 1| \tag{16}$$

## 3.6 POLICY TRAINING

We employ a policy gradient-based reinforcement learning algorithm for training the nominal controller $\pi_{optimal}$. $\pi_{adapt}$: This is mainly for data collection for training the latent space and the barrier function. $\pi_{optimal}$: The policy is entirely trained with off-policy data as the action space is modified by the LCBF. We use DDPG Lillicrap et al. (2015), as it works well with both on-policy and off-policy training. The training is divided into two phases, one with $\pi_{adapt}$ before reaching the convergence point of the latent space and the barrier function. After this, we switch to phase two with $\pi_{optimal}$ to learn a policy safely with the control barrier function. A detailed ablation on the choice of $\pi_{adapt}$ and performance of the framework with different $\pi_{optimal}$ is given in A.8.

## 4 EXPERIMENTS

### 4.1 COMPARISON TO STATE OF THE ART NEURAL CBF APPROACHES

We compare our algorithm with rCLBF-QP Dawson et al. (2022b) and BarrierNet Xiao et al. (2021) under the same experiments mentioned in these two papers for consistency. For implementing rCLBF-QP Dawson et al. (2022b), we utilized the official implementation, which is made publicly available by the authors. We also train state estimators required by the two approaches. Both approaches use predefined dynamics of the system, a predefined barrier function, and a cost function. we compare the number of episodes required to train such vision modules for state estimation to the number of episodes required for LCBF to learn the latent space and barrier function. Details of the exact experimental setup is given in A.1. We also compare the safety rate. The comparison is summarized in Table 1.

Table 1: Tracking error comparison

| Task[2] | Approach | # episodes[1] | $|x - x_{goal}|$ |
|---|---|---|---|
| Car trajectory tracking Kinematic model (only sensor data) | rCLBF-QP | 136 | 0.8621 |
| | BarrierNet | 148 | **0.6432** |
| | LatentCBF (ours) | **92** | 0.7165 |
| Car trajectory tracking Sideslip model (only sensor data) | rCLBF-QP | 132 | 0.9439 |
| | BarrierNet | 152 | **0.6957** |
| | LatentCBF (ours) | **92** | 0.7165 |
| 3D Quadrotor (only sensor data) | rCLBF-QP | 104 | 0.5321 |
| | BarrierNet | 84 | 0.6417 |
| | LatentCBF (ours) | **72** | **0.4967** |

Table 2: Safety rate comparison

| Environment[3] | Controller | # episodes[1] | Safety Rate |
|---|---|---|---|
| 2D Quadrotor with obstacles (image input of the environment) | rCLBF-QP | 224 | 78% |
| | BarrierNet | 192 | 81% |
| | LCBF | **108** | **86%** |
| 3D Quadrotor with an obstacle (image and sensor data as input) | rCLBF-QP | 272 | 98% |
| | BarrierNet | 208 | 100% |
| | LCBF | **144** | **100%** |

All the tasks are individually explained in the subsection below. We also present a Visualization of the Barrier Function learned $B(z)$, by applying NMF Lee & Seung (2000) on the observation space to embed into a 2D space corresponding to which $B(z)$ is plotted in a 3D surface. Futher details of application of NMF for visualization is present in A.2. Furthermore, the safe and unsafe samples are plotted to visualize in which region they lie according to the $B(z)$. For all the experiments training plots are available at A.7

---

[2]For trajectory tracking, we compute the maximum tracking error over the trajectory

[1]Individual epoch contains four episodes hence the number of episodes is multiple of 4.

[3]For the quadrotor, we compute % of trials reaching the goal with tolerance $\delta = 0.25$ without collision

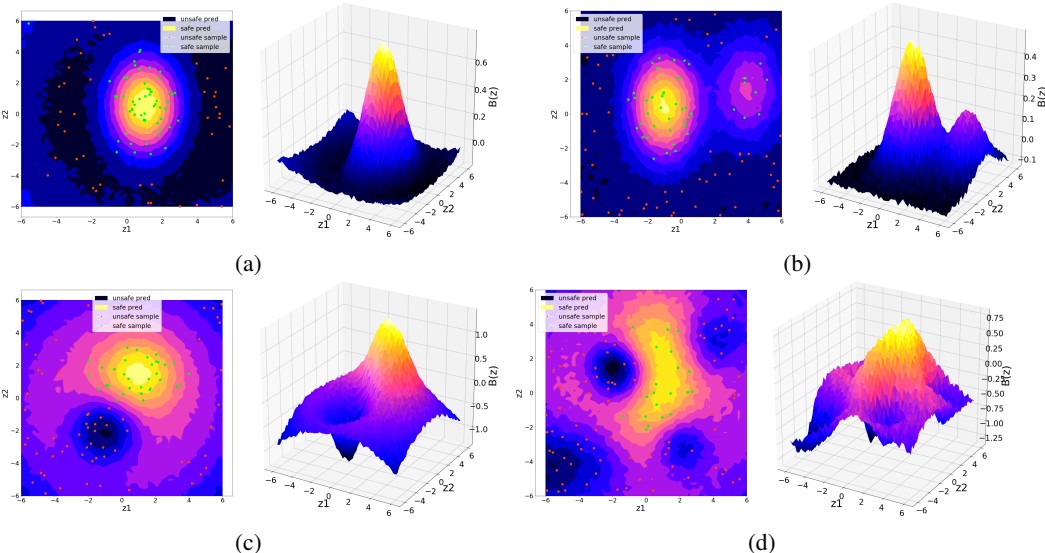

Figure 3: Visualization of $B(z)$ for the different experiments and projections of safe and unsafe samples on contour and 3D surface view. (a) Car Trajectory tracking experiment; (b) 3D quadrotor without obstacles; (c) 2D quadrotor with obstacles; (d) 3D quadrotor with obstacles

**Car Trajectory tracking**. From Dawson et al. (2022b), this task involves training different functions and matrices for rCBLF-QP and BarrierNet, aiming to trace the ego vehicle's given trajectory. BarrierNet performs best in tracking error, but our approach converges fastest, indicating a clear separation between safe and unsafe sets, as visualized in Fig. 3a. For more details on experimental setup of rCBLF-QP and BarrierNet refer Appendix. A.3

**3D Quadrotor without obstacle**. In this task, derived from Dawson et al. (2023a), our approach surpasses others in convergence and tracking error, dealing with high model dynamic uncertainty and using only sensor data. The latent space accurately represents the safe space, depicted in Fig. 3b; it is clear that the learned latent space is able to represent the safe space within the two concave surfaces. Additional experimental setup can be found in Appendix. A.4

**2D Quadrotor with obstacles**. This task involves a 2D quadcopter navigating around obstacles from Dawson et al. (2022b); Ho et al. (2020). State estimations and barrier function learning limit the system, requiring numerous iterations for convergence, but our approach still outperforms, showing well-separated safe and unsafe regions in Fig. 3c. Futher details of the task and description about the complexity of the task are in Appendix. A.5

**3D Quadrotor with obstacles** This task, set in the PyBullet gym environment, provides both ground truth and depth images as the quadrotor's observations and poses a substantial challenge due to the high dimensionality of the input space. Our approach effectively surpasses both the rCBLF-QP and BarrierNet in terms of safety rate and the required number of episodes for training. The detailed depiction in Fig. 3d of the barrier function appears non-convex but exhibits convexity in the latent space $Z$, indicated by the high correlation between the safe and unsafe samples and their corresponding regions due to the construction of $B(z)$ using a Lipschitz network. Futher details can be found in Appendix. A.6

## 4.2 LIPSCHITZ NETWORK FOR SYNTHESIS OF BARRIER FUNCTION

Our investigation illuminates the deficiencies inherent in prevalent neural barrier approaches, primarily their reliance on standard deep learning layers which neglect the incorporation of any Lipschitz criterion during the learning of barrier functions. To rectify this, we implemented Lipschitz Neural Networks (NNs) within the Autoencoder and control policy framework. Experiments were structured to facilitate a comparison between Lipschitz and standard networks, assessing their proficiency in synthesizing barrier functions within environments previously detailed, such as Car Tra-

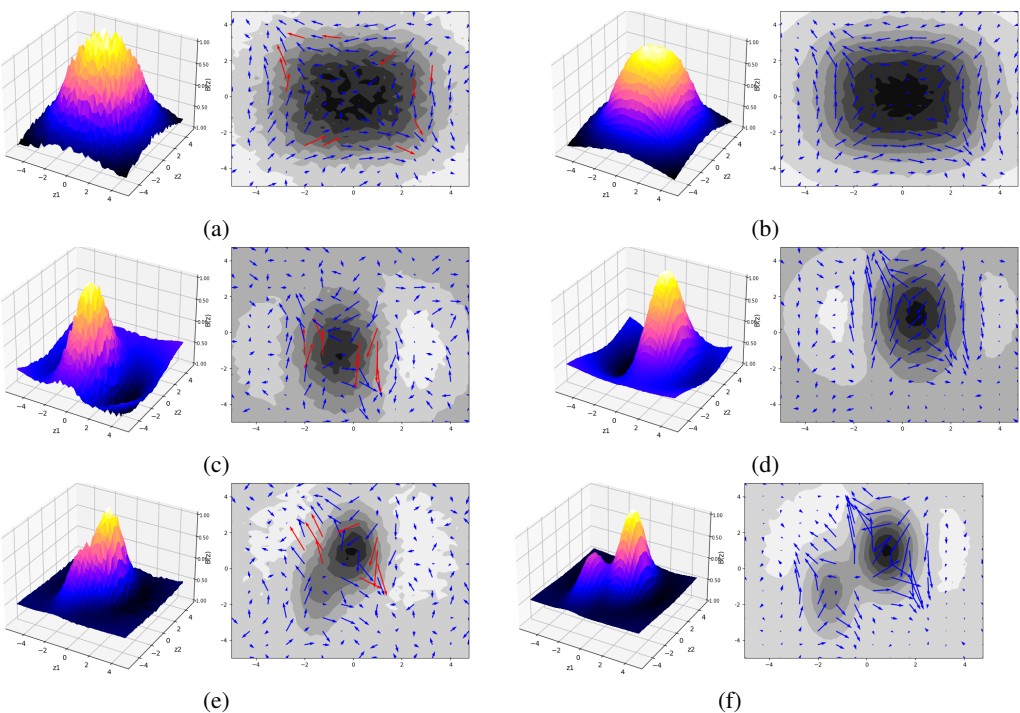

Figure 4: Visualization of $B(z)$ and $\nabla B(z)$ its gradient map for the different experiments and surface view. (a) Non Lipschitz network (b) Lipschitz network for $B(x,y) = 2^4 - (x^4 + y^4)$; (c) Non Lipschitz network (d) Lipschitz network for Car Trajectory tracking experiment; (e) Non Lipschitz network (f) Lipschitz network for 3D quadrotor without obstacles;

jectory tracking and 3D Quadrotor without obstacles Fig. 4c, Fig. 4e. Additional experiment with $B(x,y) = 2^4 - (x^4 + y^4)$ (typical CBF definition used for a cuboidal safety region across a 2D obstacle) were conducted to affirm that our framework does not restrict the synthesis of optimal barrier functions utilizing standard deep learning layers.

The barrier function $B(z)$, is pivotal for transitioning control signals or actions to safer alternatives, necessitating the computation of gradients by the QP solver from Amos & Kolter (2017). It is imperative that an optimal barrier function is differentiable and devoid of any singularities in $\nabla B(z)$. The Lipschitz criterion guarantees this absence of singularities, the effectiveness of which is visually substantiated in our results Fig. 4. Our findings reveal that standard networks encounter singularities when the value of $B(z)$ approaches zero Fig. 4a, Fig. 4c, Fig. 4e, posing a significant risk as Control Barrier Functions (CBFs) are crucial at the transition point of $B(z) \geq 0$. This underscores the relevance and appropriateness of Lipschitz Networks for our proposed framework, highlighting their role in enhancing the reliability and safety of neural barrier approaches.

## 4.3 FRAMEWORK MODULARITY ASSESSMENT

To assess the modularity of our approach, we integrated LatentCBF with existing works utilizing Encoder and Decoder architecture for complex control tasks, choosing the Carla public leaderboard CARLA Team (2020) as a benchmark against the state-of-the-art InterFuser Shao et al. (2022). Inter-Fuser uses a transformer-based architecture and employs an explicitly defined safety controller limited by various heuristics. This is suboptimal in critical scenarios as it only downsamples waypoints into a set of safe waypoints without any modification. To validate the versatility of our work with diverse tasks and architectures, we aligned our LatentCBF with the InterFuser framework, incorporating modifications like Lipschitz Transformer layers Qi et al. (2023) to maintain fair comparison. The same methodology as InterFuser paper Shao et al. (2022) was followed, with additional loss16 for LatentCBF. A comparative evaluation and an ablation study highlights enhancements brought by LatentCBF in terms of safety.

Table 3: Performance comparison on the public CARLA leaderboard CARLA Team (2020). All three metrics are higher the better. Our LatentCBF based Interfuser is better than InterFuserShao et al. (2022) in Route Completion and Infraction Score and close second for Driving score.

| Method | Driving Score | Route Completion | Infraction Score |
|---|---|---|---|
| LatentCBF based Interfuser with CBF (ours) | 74.23 | **91.21** | **0.88** |
| InterFuser Shao et al. (2022) | **76.18** | 88.23 | 0.84 |
| LatentCBF based Interfuser without CBF | 56.29 | 73.13 | 0.74 |
| Interfuser without safety controller | 28.21 | 56.67 | 0.54 |
| TCP Wu et al. (2022) | 75.14 | 85.63 | 0.87 |
| LAV Chen & Krähenbühl (2022) | 61.85 | **94.46** | 0.64 |
| TransFuser Chitta et al. (2022) | 61.18 | 86.69 | 0.71 |
| Latent TransFuser Chitta et al. (2022) | 45.20 | 66.31 | 0.72 |
| GRIAD Chekroun et al. (2023) | 36.79 | 61.85 | 0.60 |
| Rails Chen et al. (2021) | 31.37 | 57.65 | 0.56 |
| TARL Toromanoff et al. (2020) | 24.98 | 46.97 | 0.52 |
| NEAT Chitta et al. (2021) | 21.83 | 41.71 | 0.65 |

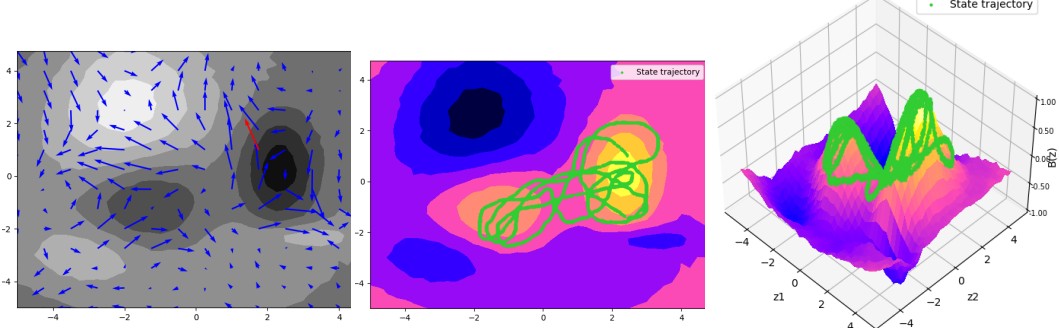

Figure 5: Visualization of $\nabla B(z)$ , $B(z)$ and projection of state trajectory into the new space. The state trajectory is an episode of latentCBF based InterFuser from Carla public leaderboard

Integrating LatentCBF for Safety Controller into InterfuserShao et al. (2022) improves Route Completion and Infraction Score over the baseline, as evidenced by 3, while maintaining comparable Driving Scores, indicating our approach's adaptability without adverse effects. The enhanced base policy, evidenced by studies foregoing CBF at inference time, diversifies the training set with critical scenarios. Visualizations of the learnt barrier function (refer to 4.2 and Fig. 5) and the gradient map, illustrate well-structured representation of safe and unsafe regions with minimal singularities, affirming suitability for CBF-QP. The trajectory visualizations in the NMF projection space further validate the efficiency of our approach in complex systems, handling transitions proficiently and ensuring trajectories remain in the $B(z) \geq 0$ region.

## 5 CONCLUSIONS

In this paper, we propose a novel framework utilizing LatentCBF, a control barrier function defined in latent space for safe control. The significance is that only observations of robots are required instead of access to the state space, which is not always available in practice. In our unified framework, we deal with AutoEncoder learning, system dynamics learning, and Barrier network learning all in latent space. Owing to the flexibility of representation learning using AutoEncoder, we expect our framework to be more general for a wide range of robotic systems. We have demonstrated the efficacy of our approach for autonomous driving benchmark in CARLA and other control affine systems like car-like robots and quadrotors. In the future, we plan to utilize latent CBF as a modular backbone to develop complex systems that could utilize the shared representation like multi-agent systems and other complex environments that lack explicit safety controller defined.

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

# A APPENDIX

## A.1 EXPERIMENTATION SETUP FOR NEURAL CBFS

We use images or a stack of images as the observation input to the encoder system, along with sensor data when available. To keep the comparison fair, we use a NN that estimates the system parameters from images and the sensor information. These networks are trained separately for the particular environment and task to act as a state estimator wherever needed by the algorithms we compare. We compare our algorithm with rCLBF-QP Dawson et al. (2022b) and BarrierNet Xiao et al. (2021) under the same experiments mentioned in these two papers for consistency. For implementing rCLBF-QP Dawson et al. (2022b), we utilized the official implementation, which is made publicly available by the authors. For the state estimation and the vision module, we employed a similar network used in Dawson et al. (2022a). For BarrierNet Xiao et al. (2021) implementation, we replicated the same system by ourselves. For the Vision model, we employed the same structure of modules as specified in Xiao et al. (2022). Both approaches use predefined dynamics of the system, a predefined barrier function, and a cost function. While these require ground truth from the system or a vision module to extract this information, we compare the number of episodes required to train such vision modules for state estimation to the number of episodes required for LCBF to learn the latent space and barrier function. We use experience from the same exploratory policy without modifying the control signal with these algorithms. Here a model is said to be converged when the test loss of these state estimators is less than a given threshold: $10^{-3}$. Our approach defined convergence when the reconstruction loss was less than $10^{-4}$.

## A.2 NON NEGATIVE MATRIX FACTORIZATION FOR VISUALING LATENT SPACE

Non-Negative Matrix Factorization (NMF)Lee & Seung (2000) is a technique widely used for dimensionality reduction and feature extraction where the data representation is exclusively non-negative. It decomposes a given non-negative matrix into two lower-dimensional non-negative matrices, rendering it particularly suitable for visualizing high-dimensional data.

### A.2.1 MATHEMATICAL FOUNDATION

Given a non-negative matrix $V$ of dimension $m \times n$, the goal of NMF is to find two non-negative matrices $W$ and $H$ of dimensions $m \times k$ and $k \times n$ respectively, where $k$ is the reduced dimensionality, such that:

$$V \approx WH$$

Here, each column of matrix $W$ can be viewed as a basis vector, and each column of matrix $H$ represents the corresponding encoding or coefficient for the basis vectors in $W$.

### A.2.2 OBJECTIVE FUNCTION

To find matrices $W$ and $H$, NMF aims to minimize the following objective function which represents the Frobenius norm of the difference between the original matrix $V$ and the approximated matrix $WH$:

$$\min_{W,H} ||V - WH||_F^2$$

subject to $W, H \geq 0$ (element-wise non-negativity).

### A.2.3 DIMENSIONALITY REDUCTION

When reducing dimensionality to two features for visualization purposes, we set $k = 2$. In this case, matrix $W$ will have the dimensionality $m \times 2$ and matrix $H$ will have the dimensionality $2 \times n$. The two columns of matrix $W$ will represent the two new features in the reduced-dimensional space, and each row of matrix $H$ will represent the encoding of the original features in the reduced-dimensional space.

Visualization is accomplished by projecting the high-dimensional data onto the two-dimensional space spanned by the two basis vectors (the columns of $W$). Each data point in the original high-dimensional space is represented as a linear combination of the basis vectors, and its coefficients in this linear combination serve as its coordinates in the two-dimensional visualization space.

In mathematical terms, for a data point represented by a column vector $v$ in matrix $V$, its corresponding two-dimensional representation is given by:

$$h = W^T v$$

Here, $h$ is a 2-dimensional vector whose components are the coordinates of the data point in the reduced-dimensional visualization space.

### A.3 EXPERIMENTAL SETUP FOR CAR TRAJECTORY TRACKING

This task, derived from Dawson et al. Dawson et al. (2022b), involved training a Lyapunov function for rCBLF-QP and training the Hessian and linear cost matrix for BarrierNet. The task is designed to trace a given trajectory to the ego vehicle. It is executed using two different vehicle dynamic models: the kinematic model with and without sideslip. From these varied experiments, consistent conclusions were derived, reinforcing the reliability of the results obtained.

### A.4 EXPERIMENTAL SETUP FOR 3D QUADROTOR WITHOUT OBSTACLE

In this task, goal tracking is performed for a 3D quadcopter Dawson et al. (2023a), with only sensor data being available. Detailed training was performed for a Lyapunov function for rCBLF-QP, cost matrices for BarrierNet, and the Latent Space. This task, characterized by high uncertainty in model dynamics, was pivotal in demonstrating the superiority of our approach in terms of convergence speed and tracking error minimization.

### A.5 EXPERIMENTAL SETUP FOR 2D QUADROTOR WITH OBSTACLES

This task is a reach-and-avoid problem for a quadcopter in a 2D space laden with obstacles Dawson et al. (2022b); Ho et al. (2020). The robot's observations are derived from a 2D image, with obstacles consistently marked with a single color. For rCBLF-QP, a system extractor was developed to locate the planar quadcopter in the 2D axis. The Lyapunov function was then trained by annotating regions with obstacles as unsafe. For BarrierNet, rectangles were superimposed for the obstacles after defining the same system extractor for location.

### A.6 EXPERIMENTAL SETUP FOR 3D QUADROTOR WITH OBSTACLES

In this task, algorithms are challenged due to the elevated dimensionality of the input space. The PyBullet gym environment is crucial as it allows the acquisition of ground truth for comparison alongside the quadrotor's observations, provided in the form of depth images. For both the rCBLF-QP and BarrierNet, a state estimator was trained meticulously to ascertain the position of the drone relative to the rounded cube object utilizing both the RGBD image and the sensor data. Subsequently, training was also conducted for the Lyapunov function for rCBLF-QP. In contrast, BarrierNet had predefined cost functions as per Xiao et al. (2021), facilitating quicker learning of the Hessian and the linear cost matrix. Our approach demonstrated superiority over both BarrierNet and rCBLF-QP controllers in safety rate and the number of episodes required for training. The barrier function revealed in Fig. 3d is notably non-convex in the embedded space from NMF, yet the high correlation between the samples and the regions indicates its convexity in the latent space $Z$, as $B(z)$ is formulated using a Lipschitz network. All baseline approaches maintained default parameters for network and optimization layers as specified in their corresponding papers.

### A.7 TRAINING PLOTS

In Fig. 6, we present the learning plots for individual components in the pipeline, i.e., the Dynamics model, Barrier Function, AutoEncoder and the policy. The grey dotted vertical line denotes the checkpoint of models we use to evaluate and get the results for the experiments given in 4.

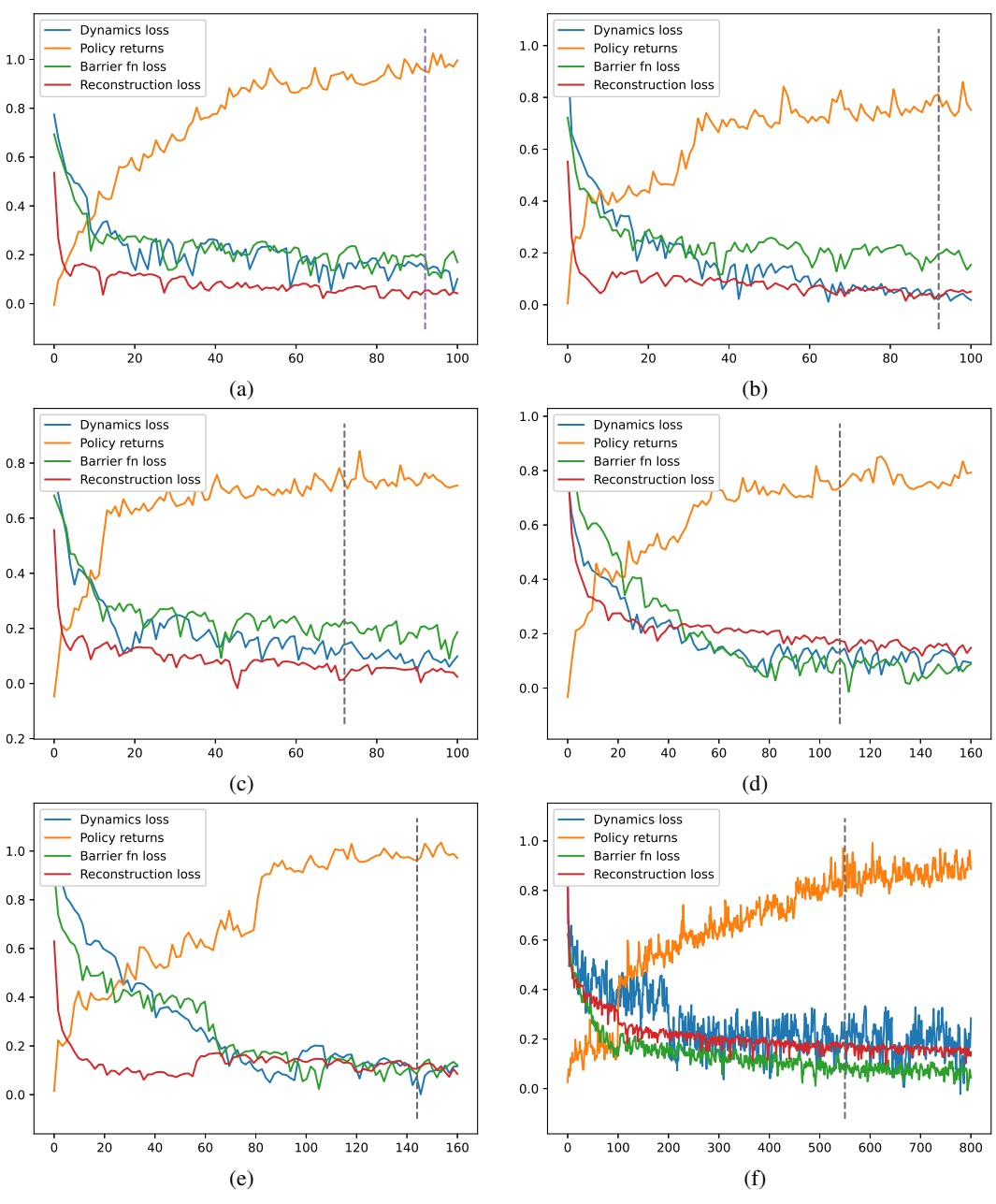

Figure 6: Training plot for the given 6 tasks with losses or returns from individual elements. x-axis is number of episodes and y-axis is the magnitude of individual metrics. (a) Car trajectory tracking Kinematic model (b) Car trajectory tracking Sideslip model (c) 3D Quadrotor (only sensor data) (d)2D Quadrotor with obstacles (e)3D Quadrotor with an obstacle (f) LatentCBF based Interfuser with CBF on Carla leaderboard

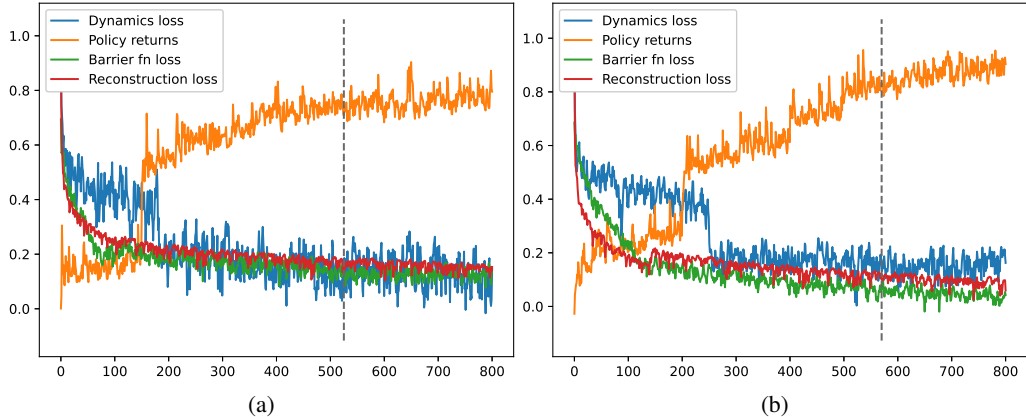

Figure 7: Training plots for the comparison of pipeline learning using $\pi_1$ and $\pi_2$ as $\pi_{adapt}$. x-axis is number of episodes, y-axis is magnitude of value of the metrics and grey dotted vertical line is the checkpoint used for evaluation.

## A.8 ABLATION STUDY OF DIFFERENT POLICIES AND CBF

To demonstrate the robustness of the presented approach, we run a few experiments to understand the choices of $\pi_{adapt}$ and $\pi_{optimal}$ and its effects in the pipeline. We divide the set of experiments into two categories of choices for 1) $\pi_{adapt}$ and 2)$\pi_{optimal}$. For being able to compare the performance, we use Carla Leaderboard as the task, with LatentCBF based Interfuser as the Pipeline.

### A.8.1 INFLUENCE OF $\pi_{adapt}$

As the process of data collection is done by $\pi_{adapt}$, this is similar to warmup training in many reinforcement learning approaches. As the samples are initially used to train the Dynamics model, Barrier Function and the AutoEncoder, it is very crucial to get sufficient samples of states that the learning policy may encounter. Hence, we ideally would like to use a policy that has the closest trajectory to the one the policy would explore while training. This is a challenging task. Hence, we demonstrate the effect of $\pi_{adapt}$ with two policies.

- **Expert Policy** $\pi_1$: For the given task of Carla Leaderboard, we have autopilot from the official Leaderboard repository that we utilize as an Expert policy.

- **Random Policy** $\pi_2$: For representing a random policy, we simply initiate a new policy and do not train it prior to or whilst data collection.

With the two policies $\pi_1$ and $\pi_2$, we have run the pipelines for the whole task. The major difference is the number of episodes required for the training with the same threshold of returns. This experiment is been carried out 10 times with different random policies for $\pi_2$. The difference in the number of episodes is $53 \pm 7$, which is much less compared to the total number of episodes to train. Hence, we simply used a random policy for tasks in 4.1 and an expert policy for 4.3. The evaluation metrics of the final checkpoints from both approaches are present in Table. 4, which shows no marginal improvement.

Table 4: Performance comparison between Our Approach with $\pi_1$ and $\pi_2$ on the public CARLA leaderboard CARLA Team (2020). All three metrics are higher the better.

| Method | Driving Score | Route Completion | Infraction Score |
|---|---|---|---|
| Our Approach with $\pi_1$ | 74.23 | 91.21 | 0.88 |
| Our Approach with $\pi_2$ | 74.12 | 91.23 | 0.88 |

### A.8.2 MODULARITY WITH $\pi_{optimal}$

For the approach to be modular, we would require the learnt CBF to work as a safety guarantee for any policy other than the one it is trained with. For the same, we experiment with two policies: 1) Jointly trained policy and 2) Policy independently trained for the same task. Hence, we compare the results of the two policies with the learnt CBF acting only when the action signal is to be modified. For the experiments, we describe the pipeline setting as below.

- **Jointly learnt policy** $\pi_3$: This is exactly the approach we use for our experiments in 4.3, a brief detail of the setup is an Expert policy was used as $\pi_{adapt}$ for training dynamics function, barrier function and the encoder, Later a random policy was trained through the proposed pipeline.

- **Independently learnt policy** $\pi_4$: As the approach above utilizes the same network architecture and policy from Interfuser Shao et al. (2022), it would be fair to train a policy from Interfuser without our pipeline elements and post-training integrate the policy into our pipeline. To do so, we utilize a separate trained encoder, dynamics model and barrier function for the same task, where the barrier function only modifies the action signal when it exits the safe set or enters the unsafe set.

With the above setting, The performance of $\pi_3$ with latent CBF pipeline, $\pi_4$ with latent CBF pipeline and $\pi_4$ without latent CBF pipeline are present in Table. 5. We see that using Latent CBF with any policy gives results on par or better and does not cause degradation of performance and hence supports the argument of modularity of the pipeline.

Table 5: Performance comparison between $\pi_3$, $\pi_4$ with and without our CBF for safety on the public CARLA leaderboard CARLA Team (2020). All three metrics are higher the better.

| Method | Driving Score | Route Completion | Infraction Score |
|---|---|---|---|
| Jointly learnt policy $\pi_3$ with LatentCBF | 74.23 | 91.21 | 0.88 |
| Independently learnt policy $\pi_4$ with LatentCBF | 76.18 | 88.23 | 0.84 |
| Independently learnt policy $\pi_4$ without LatentCBF | 75.33 | 90.26 | 0.87 |

