# OpenReview forum: "LatentCBF: A Control Barrier Function in Latent Space for Safe Control"
_ICLR.cc/2024/Conference — Submitted to ICLR 2024_

### Official Review · Reviewer_ciJn · 2023-10-25

**Soundness:** 2 fair
**Presentation:** 3 good
**Contribution:** 2 fair
**Rating:** 3
**Confidence:** 4

**Summary:**

The authors propose to learn a control barrier function in the latent space via an auto-encoder.  The system dynamics is learned by directly regressing over the gradient. The CBF is learned following the common loss setup in the CBF learning literature. The safe policy is learned by distillation from the CBF-QP result where the QP layer is differentiable following Brandon Amos's work. The proposed method is verified on a few tasks including car and quadrotor obstacle avoidance. The author also combined the latent CBF with an existing learned driving agent and achieved improvement in the Carla simulator.

**Strengths:**

1. The simulation experiment results are strong,
2. The paper is generally well-written, easy to follow
3. The idea of learning a CBF in the latent space is new

**Weaknesses:**

1. I don't get why learning a CBF in the latent space is more advantageous than learning in the original state space, and there is no comparison to show the benefit.
2. There lacks necessary theoretical "book-keeping" results to ensure that the latent CBF is well-defined and the theories for the original CBF can carry over.
3. The training process is not explained clearly, I'm confused about the training details, e.g. how is the auto-encoder trained?

**Questions:**

1. "Domain adaptation for the encoder ensuring the latent space that the latent space is always relevant and can contain features necessary for the barrier function", please remove the first "the latent space"

2. For equation (2), please specify the norm used.

3. How do you guarantee the injectiveness of the encoder, i.e., each point x got mapped to a unique z?

4. I don't think one can simply define a barrier function on z and apply the original CBF theory without any care. For example, assuming that z has a higher dimension than x (which is probably true), all z that are mapped from an x live on a manifold. The original CBF theory is built on assumptions that involve openness and compactness of certain sets, which may be violated here. For one, the image of the safe set in the latent space may even be disconnected when the original safe set is connected or the other way around.

5. I find the dynamics learning problem very weird, it seems that the assumption is that you can observe \dot{x} directly, yet the dynamics is unknown. This is almost never the case in practice as gradient observation is very noisy. The authors may consider a discrete-time CBF setting for practicality.

6. As mentioned above, please provide more details on the training process, is the training end to end, if not, how many steps and which components are fixed, which are being trained in each step?

---

> ### Author Response · Authors · 2023-11-19
> **Reply to Reviewer ciJn review's**
>
> Please find below the replies for the weakness and question mentioned by Reviewer ciJn
> - ### Weakness
> 	- I don’t get why learning a CBF in the latent space is more advantageous than learning in the original state space, and there is no comparison to show the benefit.
> 		- `The experiments comparing barrier net and rlcbf-qp are done to show why learning on latent space is advantageous in terms of increased sample efficiency with better or on-par results and experiment with Carla is demonstration scalability in high dimensional environment which would be a strenuous modelling task for previous cbf based approaches.`
> 	- There lacks necessary theoretical “book-keeping” results to ensure that the latent CBF is well-defined and the theories for the original CBF can carry over.
> 		- `We would be happy to understand what specific class of results you are trying to point.`
> 	- The training process is not explained clearly, I’m confused about the training details, e.g. how is the auto-encoder trained?
> 		- `We apologize for the same, we have trained the autoencoder with (9), (14) and (16) losses on the samples from \pi_{adapt}.`
> - ### Questions
> 	- “Domain adaptation for the encoder ensuring the latent space that the latent space is always relevant and can contain features necessary for the barrier function”, please remove the first “the latent space”
> 		- `Thank you for pointing out will do the same.`
> 	- For equation (2), please specify the norm used.
> 		- `Equation (2) is for denoting the basic Lipschitz criterion, for our use its Euclidean Norm.`
> 	- How do you guarantee the injectiveness of the encoder, i.e., each point x got mapped to a unique z?
> 		- `As we aim to learn a compact but rich representation, we do not specifically check for a one-to-one behaviour of our encoder as we need it to generalize over different conditions while only being able to represent important features required by the policy and the CBF.`
> 	- I don’t think one can simply define a barrier function on z and apply the original CBF theory without any care. For example, assuming that z has a higher dimension than x (which is probably true), all z that are mapped from an x live on a manifold. ... For one, the image of the safe set in the latent space may even be disconnected when the original safe set is connected or the other way around.
> 		- `We agree that for high dimensional data like the one we present here containing information from images and sensors, the final representation learnt may have many finite subcovers. However, we have visually demonstrated that the subcovers for the safe set are well separated on the manifold. It is important to understand that the 2-dimensional space on which the manifold is plotted is obtained from NMF of the latent space, which leads to the union of these subcovers into the visualized manifold. The crucial implementation detail which assists the formation of closed sets for defining a barrier function according to the theory is Lipschitz networks as opposed to simple neural network layers and activations. Reducing the number of closed sets that are safe sets. This is an approximation for designing our system and not a guarantee.`
> 	- I find the dynamics learning problem very weird, it seems that the assumption is that you can observe \dot{x} directly, ... The authors may consider a discrete-time CBF setting for practicality.
> 		-  `We agree that Discrete-time analysis would be better for defining System dynamics; while this would be a good direction of experimentations, this would detract from the current approach as the discrete control barrier function involves multiple challenges. The approach of learning the dynamics via gradient observation is a good approximation, gradient observation has been employed in multiple works (9) in [1], (27) in [2], (25) in [3], where such gradients are used as ground truth or have been part of the loss function for uncertain dynamics models. References: [1] Learning for Safety-Critical Control with Control Barrier Functions [2] Online Adaptive Compensation for Model Uncertainty Using Extreme Learning Machine-based Control Barrier Functions [3] Learning a Better Control Barrier Function Under Uncertain Dynamics`
> 	- As mentioned above, please provide more details on the training process, is the training end to end, if not, how many steps and which components are fixed, which are being trained in each step?
> 		- `We will update the manuscript with better details of the training process. There are two phases of training: one for sample collection done for training the dynamics and the autoencoder, similar to system identification with \pi_adapt Policy; this is a policy with no safety guarantees, and phase 2 is training of policy \optimal (which is the same policy \pi_adapt), CBF and autoencoder all the results mentioning the number of episodes is the sum of episodes from both the phases.`

---

> > ### Comment · Reviewer_ciJn · 2023-11-20
> >
> > I appreciate the authors providing additional details and explanations. However, my main concern remains,  that is, this paper doesn't provide much theoretical advancement for neural CBF, and on the practical side, the improvement over existing learning-based CBF methods is not significant. Overall, I find the paper's contribution below the bar of ICLR.

---

> ### Author Response · Authors · 2023-11-21
> **Reply to Reviewer ciJn**
>
> Thank you for your elucidative comments about the work. We do agree that the paper doesn't contribute to any theoretical advancement of CBFs but rather a modular pipeline for working with partially observed systems in RL. We would be happy to discuss which other learning-based CBF methods are suitable for such a task with unknown system definitions.

---

### Official Review · Reviewer_ERPL · 2023-10-28

**Soundness:** 2 fair
**Presentation:** 3 good
**Contribution:** 2 fair
**Rating:** 5
**Confidence:** 4

**Summary:**

This work uses autoencoder to learn latent space within which the dynamics and CBF are well-defined and proposes an algorithm for learning dynamics/CBF and an optimal policy together; which is applied to some practical tasks showing advantages.

**Strengths:**

1. For some experiments, it is shown that the proposed algorithm successfully learn CBF approximately with autoencoder architectures.
2. Algorithms are simple enough.

**Weaknesses:**

1. There are some mathematically no rigorous or strange writings; e.g., page 3 top “are both in R^D” should be “are both in the Euclidean space R^D” or so.  Also, B(z, theta_B) should be continuously differentiable?  And B(z, theta_B) should be on R^d x parameter space (or theta_B could be given…?)?  Eq 16 is a bit strange; z in Z_safe within dataset?
2.  If you assume the dynamics is over the observation space (or learned latent space), it has all the information about the state; so I get the idea that the authors are trying to do encoding to get latent state, the work is not dealing with the partial observability case.  Abstract and introduction are misleading (no access to “all states” may imply that we have partial observability).
And it is mathematically just CBF (not LatentCBF or so).
3.  CBF is useful for giving theoretical guarantees of safety.  Although adding encoding/decoding layers for practical purpose, it adds little insights about the theoretical concept of CBF.

**Questions:**

1. What is the systematic and robust choice of policy_adapt?  If the whole algorithm, including the learning of CBF, is robust, I believe the work becomes solid.  For now, it seems a bit ad-hoc, which hinders the benefits of practical applications (as I mentioned, the focus of this work is practical scaling to image space etc. rather than theoretical insights.  So each choice of parameters, exploration policies etc. should be made clear and robust.

---

> ### Author Response · Authors · 2023-11-19
> **Reply to Reviewer ERPL review's**
>
> Please find below the replies for the weakness and question mentioned by Reviewer ERPL
> - ### Weakness
> - There are some mathematically no rigorous or strange writings; e.g., page 3 top “are both in R^D” should be “are both in the Euclidean space R^D” or so. Also, B(z, theta_B) should be continuously differentiable? And B(z, theta_B) should be on R^d x parameter space (or theta_B could be given…?)? Eq 16 is a bit strange; z in Z_safe within dataset?
> 	- `Thanks for pointing out the errors in writing, we will fix them.`
> - If you assume the dynamics is over the observation space (or learned latent space), it has all the information about the state; so I get the idea that the authors are trying to do encoding to get latent state, the work is not dealing with the partial observability case. Abstract and introduction are misleading (no access to “all states” may imply that we have partial observability). And it is mathematically just CBF (not LatentCBF or so).
> 	- `We agree we are not solving partial observability directly, while our approach is only to use the observation from sensors and vision modules. In contrast, implicit definitions of CBF use state information, such as the position of obstacles and soon, which is not directly accessible in practical scenarios. Mathematically, it is simply a CBF as it is a synthesis of a CBF which can update based on chosen reward and environment.`
> - CBF is useful for giving theoretical guarantees of safety. Although adding encoding/decoding layers for practical purpose, it adds little insights about the theoretical concept of CBF.
> 	- `As rightly outlined, our work focuses on practical usage of CBF as a generalized module for deep reinforcement learning approaches.`
>
> - ### Questions
>
> - What is the systematic and robust choice of policy_adapt? If the whole algorithm, including the learning of CBF, is robust, I believe the work becomes solid. For now, it seems a bit ad-hoc, which hinders the benefits of practical applications (as I mentioned, the focus of this work is practical scaling to image space etc. rather than theoretical insights. So each choice of parameters, exploration policies etc. should be made clear and robust.
> 	- `Thank you for pointing out the ambiguity in the selection decisions of such hyperparameters. We are in progress of updating the current manuscript with an addition section in the Appendix for the same.`

---

> > ### Comment · Reviewer_ERPL · 2023-11-20
> > **Thank you for the response**
> >
> > Thank you for the response to my concerns.
> > I wanted to let you know that I have read your responses.
> >
> > For the first part of Weakness, i.e., about mathematical rigors, I put some questionings rather than corrections.
> > So it would be great if you could answer to those questions.
> > Also please let me know when you update the part about the systematic and robust choice of parameters; if it can be clear from the response text, it is fine with me as well (no need to thoroughly update the manuscript at this time)

---

> > > ### Author Response · Authors · 2023-11-21
> > > **Reply to Reviewer ERPL review's**
> > >
> > > We have updated the analysis of $\pi_adapt$ selection and also commented about the works modularity for newer policies.
> > >
> > > Answering your original question, we missed
> > > - There are some mathematically no rigorous or strange writings; e.g., page 3 top “are both in R^D” should be “are both in the Euclidean space R^D” or so. Also, B(z, theta_B) should be continuously differentiable? And B(z, theta_B) should be on R^d x parameter space (or theta_B could be given…?)? Eq 16 is a bit strange; z in Z_safe within dataset?
> > >    - `We have fixed the first part, which was an error in writing. Yes, B(z, theta_b) is required to be continuously differentiable over the trajectory for the optimization problem defined in Eq 6. This is also required for non-singular solutions of the optimization problem.  Yes, it is also defined on the latent space, and the shape of theta_b is d x 1, 1 because output of barrier function is just a value`
> > >
> > > In Eq 16, $z \in Z_{safe}$ under the summation means we are doing a summation for all the elements in $Z_{safe}$, which we get from a set defined in Eq 15 on the trajectory collected.

---

> > > > ### Comment · Reviewer_ERPL · 2023-11-22
> > > > **Thank you again for your response**
> > > >
> > > > Thank you again for your responses; although I keep my score, I think it will be an important work if the practicality of this work (rather than the CBF theory insights) with more guidance on how one could use it to diverse domains is studied well.

---

### Official Review · Reviewer_pAha · 2023-10-28

**Soundness:** 3 good
**Presentation:** 4 excellent
**Contribution:** 2 fair
**Rating:** 5
**Confidence:** 4

**Summary:**

Traditional control methods for enforcing set-invariant safety involve the construction of a scalar functional such as a CBF to ensure projection to safe controls. With the advent of learning, several models that were used in generating safe controls have been replaced with learned models such as neural nets. For example, several papers learn a neural CBF while others try to fit a NN to both CBF and unknown dynamics. Here, the authors endeavor to relax the assumption of a known system/robot state. Rather, only a high-dimensional observation such as an image is known. The projection to latent space is performed using an auto-encoder to generate a latent representation. This representation is a proxy for the state and the CBF/dynamics is learnt over the representation. To learn the CBF, an appropriate loss function to separate safe and unsafe trajectories is used. The latent representation is learned end-to-end with an appropriate auto-encoder loss. The dynamics is also modeled with a neural net and learned during an exploration phase. Experimental validations are performed on several benchmarks common in CBF literature such as the quadrotor and kinematic car dynamics. Additional validation is performed using the CARLA simulator.

**Strengths:**

1) The novelty is clear and is a good fit for this venue.

2) The experimental results and comparisons are good. The result in CARLA shows some additional effort on the authors' side.

3) The presentation and the figures in this paper are excellent and demonstrate the results clearly.

4) The specific use of Lipschitz neural networks and Bjorck layers is interesting.

**Weaknesses:**

Some of the weaknesses in terms of significance of the results and related questions are listed here.

1) While it seems good to assume no knowledge of any aspect of the dynamics/environment, the solution can retain some structural aspects of the dynamics/environment. The approach here is to learn everything as a black box.  In contrast, this paper ( https://arxiv.org/pdf/2203.02401.pdf  ) uses whatever ego-state information is available as proxy intermediate losses while training end-to-end. That might be a more realistic scenario where we have some partial information available.

2) It would be interesting for me to understand if the training is stable. In terms of the training, it would be nice to see learning curves and error bars as to how robust and repeatable the training is.

3) Some of the appeal of classical control is the existence of provable guarantees especially when it comes to safety (even though those guarantees come with strings attached). While ensuring a self-driving car is safe, we are not happy if the car is safe $x\\%$ of the time. We are looking for something stronger. In this context, apart from the percentage of time the system is unsafe, it would be nice to understand how much the deviation from the safe boundary is.

4) Safety is not guaranteed during the exploration phase

**Questions:**

1) “The major limitation is that the typical definition of a barrier function requires state information that is not generally available in real-world scenarios.” - Usually some partial state information is available so the premise is not perfectly true.

2) When the CBF is learnt using a soft loss function, it is not for theoretical purposes a strict CBF. Rather, it is an approximate neural CBF.  Is there any way of overcoming this limitation such as using verification?

3) For the baseline comparison, is the state-assumed to be known or estimated with a different neural module. More clearly, is the main difference between the current method and baselines mainly end-to-end learning?

4) It is interesting that LCBF uses fewer episodes for learning. Is it because all components of the state are not relevant to safety and thus need not be learnt in the latent state?

5) Will the method work if $\pi^{optimal}$ changes?

---

> ### Author Response · Authors · 2023-11-19
> **Reply to Reviewer pAha review's**
>
> Please find below the replies for the weakness and question mentioned by Reviewer pAha
> - ### Weakness:
> - While it seems good to assume no knowledge of any aspect of ... That might be a more realistic scenario where we have some partial information available.
> 	- `The major limitation of the approach the reviewer refers to is access to true state value at training time, while the approach does not assume any such assumption and is able to train on tasks in Carla leaderboard benchmark.`
> - It would be interesting for me to understand if the training is stable. In terms of the training, it would be nice to see learning curves and error bars as to how robust and repeatable the training is.
> 	- `Sure, we will add training plots for dynamics, encoder and policy for selected experiments.`
> - Some of the appeal of classical control is the existence of provable guarantees especially when it comes to safety (even though those guarantees come with strings attached). ...  it would be nice to understand how much the deviation from the safe boundary is.
> 	- `While we have tried to submit the standard metrics with each environment, the CARLA leaderboard benchmark does not have any golden benchmark trajectory to compare our final policy performance with. We have produced |x_{traj} - x_{goal}| for the other environments. The computation of time to recover from an unsafe state would be highly biased as CBF is synthesized and not defined, leading to approximate results. `
> - Safety is not guaranteed during the exploration phase
> 	- `This is true and also a quintessential step for training. In Carla Leaderboard, the Route Completion is lower than others because of robustness as \pi_adapt is an MPC that rarely traverses through the unsafe set. Hence, exploration of safe and unsafe becomes crucial to learning a robust CBF.`
> - ### Questions:
> - “The major limitation is that the typical definition of a barrier function requires state information that is not generally available in real-world scenarios.” - Usually some partial state information is available so the premise is not perfectly true.
> 	-  `I believe the reviewer wants to address the partial availability of state information. Hence, we would like to point out that classical definitions of barrier function uses state information that is not directly available from sensors or vision modules and defining a barrier function on the partial state information is very challenging given vision and lidar information of the used environments. Hence, appealing synthesizing CBFs for such use cases`
> - When the CBF is learnt using a soft loss function, it is not for theoretical purposes a strict CBF. Rather, it is an approximate neural CBF. Is there any way of overcoming this limitation such as using verification?
> 	- `Verification of such an approach would be a great contribution to this work, but verification poses multiple challenges to understanding a baseline CBF for complex systems in latent or the Original space. Hence, we have only evaluated the performance of the system and visualized its trajectories.`
> - For the baseline comparison, is the state-assumed to be known or estimated with a different neural module. More clearly, is the main difference between the current method and baselines mainly end-to-end learning?
> 	- `While we have tried to explain the difference between the baselines used and our approaches at the start of 4.1, summarizing the following is that other baseline approaches require defined dynamics and cost matrices and even require the use of an external state estimator for the tasks that require it. Our approach does not explicitly require any of these definitions. Hence, easier end-to-end learning approach for alike tasks.`
> - It is interesting that LCBF uses fewer episodes for learning. Is it because all components of the state are not relevant to safety and thus need not be learnt in the latent state?
> 	- `The efficiency of LCBF is higher than that of the other approaches primarily because of joint training of the latent space to optimize the loss for improving safety definition and the Policy while having all relevant state information through the reconstruction loss. This makes the system hyperparameter sensitive on the weights of individual losses for the encoder, But with sufficient experimentation, it is straightforward to get a balance of individual losses.`
> - Will the method work if \pi_optimal changes?
> 	- `I Believe you have misunderstood \pi_optimal with \pi_adapt, as \pi_optimal is the policy that is learnt through training. And \pi_adapt is used for training sample collection. Yes, the policy \pi_adapt plays a critical role in the robustness of the learnt CBF. As can be seen in CARLA experimentation. A sub-optimal policy which can explore the environment is sufficient for the task, and hence, we have not added additional information for the same.`

---

> > ### Comment · Reviewer_pAha · 2023-11-21
> > **Thank you for the response**
> >
> > 1) In RL, the training is always done in simulation and it should be okay to have access to state at training time. I would not dwell on this weakness at this stage too much. It is something to think about.
> >
> > 2) I agree with the authors that exploration is important for learning.
> >
> > 3) "I believe the reviewer wants to address the partial availability of state information. Hence, we would like to point out that classical definitions of barrier function uses state information that is not directly available from sensors or vision modules and defining a barrier function on the partial state information is very challenging given vision and lidar information of the used environments. Hence, appealing synthesizing CBFs for such use cases" - I am referring to the output-feedback control problem raised by Reviewer 5PWg. [A1] is making a contribution towards solving this problem.
> >
> > 4) Given that $\pi_{optimal}$ is jointly trained, I am referring to the problem of distribution shift or covariate shift. If a different policy that is not jointly trained is used, will this setup work out-of-the-box like a classical CBF?
> >
> > [A1] Dean, Sarah, et al. "Guaranteeing safety of learned perception modules via measurement-robust control barrier functions." Conference on Robot Learning. PMLR, 2021.

---

> ### Author Response · Authors · 2023-11-21
> **Reply to Reviewer pAha review's**
>
> Reviews
> 1. In RL, the training is always done in simulation and it should be okay to have access to state at training time. I would not dwell on this weakness at this stage too much. It is something to think about.
> 	1. `Majorly we wanted to focus on algorithms that can be used in realworld, and assumption from simulation make it difficult to transfer algorithms from sim to real, https://arxiv.org/pdf/2203.02401.pdf is a good example on how they extended the work https://arxiv.org/pdf/2111.11277.pdf for real world system using state estimators. Making the pipeline highly depending on these state estimators which could fail in multiple scenarios and hence propagating errors. We surely dont address the problem of sim-2-real transfer but wanted to follow this goal.`
>
> 2. I agree with the authors that exploration is important for learning.
>
> 3. "I believe the reviewer wants to address the partial availability of state information. Hence, we would like to point out that classical definitions of barrier function uses state information that is not directly available from sensors or vision modules and defining a barrier function on the partial state information is very challenging given vision and lidar information of the used environments. Hence, appealing synthesizing CBFs for such use cases" - I am referring to the output-feedback control problem raised by Reviewer 5PWg. [A1] is making a contribution towards solving this problem.
> 	1. `Right I believe i understand the question better now, Yes we are trying to solve the same problem as mentioned in the reply to 5PWg. [A1] is not output feedback control as they use measurement module that are perception modules to estimate the state space for the original dynamics of the problem. They are trying to unify techniques to compensate for errors in measurement modules by formulating it as a cbf for unknown dynamics. Our approach is much different as no work has explored cbf for output feedback control.`
>
> 4. Given that $\pi_{optimal}$ is jointly trained, I am referring to the problem of distribution shift or covariate shift. If a different policy that is not jointly trained is used, will this setup work out-of-the-box like a classical CBF?
> 	1. `We have addressed the issue in the new rebuttal revision of the paper please let us know if you require any clarification on it.`

---

> > ### Comment · Reviewer_pAha · 2023-11-23
> >
> > Thank you for the clarification. Overall, I believe I provided fair scores to this paper and I would like to retain my score.
> >
> > If the method is geared towards training in the real-world, it needs to be safe during exploration.
> >
> > Estimating the state from measurements is exactly one way of output feedback using what is known as the separation principle.

---

### Official Review · Reviewer_5PWg · 2023-10-29

**Soundness:** 2 fair
**Presentation:** 1 poor
**Contribution:** 2 fair
**Rating:** 3
**Confidence:** 4

**Summary:**

This paper addresses the problem of safe data-driven control using neural networks when access to the state is unavailable. Observations (not necessarily the true states) are embedded in a latent space using a Lipschitz autoencoder. Using this latent representation of the dynamics, a QP-CLF based approach is used to design a safe controller in the latent space, by learning the dynamics in latent space, the controller and the barrier certificate. The safe and unsafe sets themselves are not explicitly defined; rather, they are annotated in the data using a simple formulation (eq 15). The empirical evaluation shows that the proposed approach achieves performance comparable to the state of the art.

**Strengths:**

The paper has the following strengths.

* The idea of embedding the dynamics in a latent space, and performing the safe control optimization therein is an interesting one, as it enables learning of the dynamics as well as the safe controller.
* In this formulation, neither the safe set nor the dynamics need to be specified and can be learned from data.

**Weaknesses:**

The paper has several weaknesses.
* The discussion about safety and barrier certificates feels rushed. Safety, as defined in [1],[2], is always with respect to safe and unsafe sets. The barrier certificate conditions given in equations (4)-(5) should hold on the interior of the safe set (see, for instance, equations (4), (7) in [1]).
* There are no discussions about guarantees that the closed-loop trajectories in the latent space will satisfy safety constraints in the canonical space (i.e. the space X from which trajectory data is initially drawn from).
* Similarly, the learning problem should be more clearly stated - what is the dataset in question, what are the functions we are learning, and what is the loss function?
* There is very limited discussion as to why Lipschitz autoencoders are necessary. In section 3.3, it's stated that this choice was made owing to [1], which requires the barrier certificate $B(z)$ to be Lipschitz. This is a little imprecise, as all the theorems in [1] only require that the barrier certificate (and the various functions, such as the state-dependent Hessian $H(x)$ used in the QP) be locally lipschitz.
* There is limited discussion about other data-driven control methods, including reinforcement learning, kernel methods, gaussian processes, etc. What are the particular advantages of using potentially costly neural policies over other methods?
* It is unclear what the authors mean with the third contribution - could that be clarified? It is not discussed anywhere else in the paper.






[1]  *Control barrier function based
quadratic programs for safety critical systems.* Ames et al, 2016.

[2] *A framework for worst-case and stochastic safety verification using barrier certificates*. Prajna and Rantzer, 2007.

**Questions:**

I ask that the authors please address the points raised in the 'Weaknesses' section. Additionallly, I have the following questions for clarification.

* It's stated in the paper that this work is motivated (at least in part) by the fact that the true state may not be accessible. This problem can be thought of as a variant of output feedback control, which has a rich history. Can you comment on the similarities and differences between this work, and other work that addresses the problem of data-driven output feedback control?
* Following the previous question, can one think of the latent-space embedding as a form of neural observer? If not, why not?
* Is it possible to formally analyze the performance of the closed-loop trajectories once they are decoded? For instance, is it possible to analyze the probability that the closed-loop trajectories are safe?
* Is there a practical advantage of using this technique in real-world scenarios? In particular, can you comment on the cost of inference using this neural network-based approach, to other methods?

---

> ### Author Response · Authors · 2023-11-19
> **Reply to Reviewer 5PWg review's**
>
> Please find below the replies for the weaknesses mentioned by Reviewer 5PWg
> - ### Weakness
> 	- The discussion about safety and barrier certificates feels rushed. Safety, as defined in [1],[2], is always with respect to safe and unsafe sets. The barrier certificate conditions given in equations (4)-(5) should hold on the interior of the safe set (see, for instance, equations (4), (7) in [1]).
> 		- `The original definition of control barrier function (7) and function for safe set in (4) are valid. We follow a more relaxed and application-oriented definition of CBF from [3].`
> 		[3]Robust Control Barrier–Value Functions for Safety-Critical Control`
> 	- There are no discussions about guarantees that the closed-loop trajectories in the latent space will satisfy safety constraints in the canonical space (i.e. the space X from which trajectory data is initially drawn from).
> 		- `Can you clarify the necessity of defining a safety constraint in canonical space? To how much we understand the question, the argument we would like to put forward is that the encoder is only a mapping from canonical space to latent space. By using latent space for representing the manifold, it is still just a projection of the Canonical space. We evaluate the state of the agent from the environment for measuring safety metrics; hence, the results are supposed to be used to evaluate the safety guarantees.`
> 	- Similarly, the learning problem should be more clearly stated - what is the dataset in question, what are the functions we are learning, and what is the loss function?
> 		- `We will surely make it clearer in the revised manuscript, and the straightaway answer is that the functions learned are Barrier Function, Dynamics, Policy, Encoder, Hessian and linear cost matrices. Losses for Barrier Function is given by (16); for Dynamics, it is (9); for Policy, it is from the algorithm you choose. For Hessian and Linear cost matrices, it is (14). For the encoder, it is the aggregation of all the above losses. The dataset in question is episodes from each respective environment. The training process is split in two, one for data collection \pi_adapt where labelling of states in these trajectories is done through (15). Later, the Policy with safety guarantees is trained, i.e. \pi_optimal, while these trajectories are also used to update other algorithms.`
> 	- There is very limited discussion as to why Lipschitz autoencoders are necessary. In section 3.3, it’s stated that this choice was made owing to [1], which requires the barrier certificate to be Lipschitz. This is a little imprecise, as all the theorems in [1] only require that the barrier certificate (and the various functions, such as the state-dependent Hessian  used in the QP) be locally lipschitz.
> 		- `For neural networks to be locally Lipschitz requires much finetuning[1]. Our functions, even though they use Lipschitz autoencoder they are locally Lipschitz with the bjork layers as through the training they only see a subset of the true state space which comes from exploration.`
> 			[1] Exactly Computing the Local Lipschitz Constant of ReLU networks
> 	-  There is limited discussion about other data-driven control methods, including reinforcement learning, kernel methods, gaussian processes, etc. What are the particular advantages of using potentially costly neural policies over other methods?
> 		- `The work majorly focuses on end-to-end learning; on the contrary, data-driven control barrier functions may perform better, but they require prior modelling and knowledge of the system. The crucial idea of using neural policies over standard control algorithms is again the generalization offered in real-world scenarios. Hence, in the related works, we have majorly cited works that aim to solve similar goals.`

---

> > ### Comment · Reviewer_5PWg · 2023-11-20
> > **Response to Authors**
> >
> > I thank the authors for their response. Several of my questions have been answered. However, some issues, particularly concerning the first question raised, have not been satisfactorily addressed. Specifically:
> >
> > * The authors state that their formulation follows that in [3]. This does not seem to be the case - in [3], the safe set is explicitly defined in Equation (2), is stated clearly in the Problem Formulation, Definition 3, and in other places as well.  Safety is always with respect to safe and unsafe sets, and must be defined as such
> > * To the authors second response, is it possible to prove (or at least demonstrate empirically) that trajectories that are safe/stable in the latent space will remain safe/stable in the original space? If not, this is a serious drawback of this work, as there would be no guarantee that the method would work once deployed. Formally speaking, suppose $\mathbb{R}^n$ is the original space, and suppose $\mathcal{C}\subset\mathbb{R}^n$ is the safe set. Let $\mathcal{Z}$ be the latent space. How does one relate safety with respect to $\mathcal{C}$ to trajectories in $\mathcal{Z}$?

---

> > > ### Author Response · Authors · 2023-11-21
> > > **Reply: Response to Authors**
> > >
> > > - Sorry for misunderstanding your question, Equation(2) in [3] defines a *safety target function* which is being learnt in our paper. Let us assume $Z_{safe} \in \set{ z: h(z) \geq 0 } $ defines the safe set and $Z_{unsafe} \in \set{ z : h(z) < 0 }$ define unsafe set. We have $B(z)$ in place of $h(z)$; hence we use (16) to optimize $B(z)$. This is the reason why we did not write the definition of $Z_{safe}$ or $Z_{unsafe}$ sets and only used the definitions for labelling samples using (15).
> > > - If we already have a safe set defined $\mathcal{C} \in \mathbb{R}^D$ we can use the Decoder network in (1), which is trained with $L_{recon}$ in our formulation. This can be done by taking a trajectory in z (latent space) and decoding back to the Original Space where one can verify with the safe set $\mathcal{C}$.
> > > $$
> > > z_{traj} \sim \mathcal{Z}
> > > $$
> > > $$
> > > \hat{x_{traj}} = D(z_{traj})
> > > $$
> > > $\hat{x_{traj}} \in \mathcal{C}$ for trajectory in safe set. \\\
> > > $\hat{x_{traj}} \notin \mathcal{C}$ for trajectory in unsafe set.

---

> > > > ### Comment · Reviewer_5PWg · 2023-11-23
> > > > **Response to Authors**
> > > >
> > > > I'd like to thank the authors for their continued discussion. However, while some of the ideas proposed in this work are interesting, I feel there are still several areas that lack clarity, particularly with respect to the safety set (the implicit safety constraints, and the injectivity of the mapping between the latent and canonical spaces). Based on this, I do not feel I can raise my score at this time.

---

> ### Author Response · Authors · 2023-11-19
> **Reply to Reviewer 5PWg review's**
>
> Please find below the replies for the questions mentioned by Reviewer 5PWg
> - ### Questions
> 	- It’s stated in the paper that this work is motivated (at least in part) by the fact that the true state may not be accessible. This problem can be thought of as a variant of output feedback control, which has a rich history. Can you comment on the similarities and differences between this work, and other work that addresses the problem of data-driven output feedback control?
> 		- `Yes, one can view the problem as a simple data-driven output feedback control for a nonlinear system. The only difference is the lack of safety guarantees from a Barrier Function. The rest of the system with Encoder, Policy and the Learnt Dynamics can be seen as a data-driven output feedback controller. To the best of our knowledge, a Control Barrier Function for such a controller has not been formulated and hence have not compared such approaches.`
> 	- Following the previous question, can one think of the latent-space embedding as a form of neural observer? If not, why not?
> 		- `It may seem the encoder is acting as a neural observer, but unlike neural observer, the dynamics are not fixed and are also being learned for the new latent space from the encoder. Hence, we see it from a perspective of joint learning of system dynamics, policy and the CBF for the latent space and iteratively updating the latent space representation.`
> 	- Is it possible to formally analyze the performance of the closed-loop trajectories once they are decoded? For instance, is it possible to analyze the probability that the closed-loop trajectories are safe?
> 		- `Yes, after decoding the trajectory from latent space to original space or using any closed loop trajectory from any other policy or controller can be verified with the learnt barrier function along with the encoder, similar to using CBF for certificates.`
> 	- Is there a practical advantage of using this technique in real-world scenarios? In particular, can you comment on the cost of inference using this neural network-based approach, to other methods?
> 		- `The cost of inference of the system would surely be higher compared to fundamental approaches for individual components of the system, while it would also not be a fair comparison due to different hardware acceleration schemas traditional approaches use.`

---

### Author Response · Authors · 2023-11-21
**Rebuttal Revision**

Thank you to all reviewers for taking the time to give constructive feedback. We have added the necessary changes to the paper. Please go through them.
Major changes is
- removing contribution 3
- fixing all the mathematical rigours
- Ablation study to understand the importance of choosing $\pi_{adapt}$ and how one would go about choosing the same. Appendix 8.1
- Modularity of the approach in terms of new non-jointly trained $\pi_{optimal}$. Appendix 8.2
- Training plots of the pipeline to understand the stability of the training process. Appendix 7

We greatly appreciate to reviewing our manuscript and re-evaluating the score of our paper! We would like to hear your questions and answer them at the earliest.

---

### Meta-Review · Area_Chair_X22b · 2023-12-14

**Metareview:**

This paper investigate the problem of safe control in partially observable state conditions. The idea of embedding the dynamics in a latent space, and performing the safe control optimization therein is an interesting one, as it enables learning of the dynamics as well as the safe controller.

There is a lack of guarantees that the closed-loop trajectories in the latent space will satisfy safety constraints in the original space. More discussion and comparison with other data-driven control methods with control barrier functions would be appreciated.

**Justification For Why Not Higher Score:**

The meta reviewer read the paper and discussions. The meta reviewer agrees with all the reviewers that this paper hasn't reached the bar of ICLR 2024.

**Justification For Why Not Lower Score:**

N/A

---

### Decision · Program_Chairs · 2024-01-16

Reject